



# SELF v1.0: A minimal physical model for predicting time of freeze-up in lakes

Marco Toffolon[1], Luca Cortese[1,2], and Damien Bouffard[2]

[1]Department of Civil, Environmental and Mechanical Engineering, University of Trento, Italy
[2]Eawag, Swiss Federal Institute of Aquatic Sciences, Department Surface Waters Research & Management, Kastanienbaum, Switzerland

**Correspondence:** Marco Toffolon (marco.toffolon@unitn.it), Damien Bouffard (Damien.Bouffard@eawag.ch)

**Abstract.** Predicting the freezing time in lakes is pursued by means of complex mechanistic models or by simplified statistical regressions considering integral quantities. Here, we propose a minimal model (SELF) built on sound physical grounds, which focuses on the pre-freezing period that, in dimictic lakes, goes from mixed conditions (lake temperature at 4°C) to the formation of ice (0°C at the surface). The model is based on the energy balance involving the two main processes governing the inverse stratification dynamics: cooling of water due to heat loss and wind-driven mixing of the surface layer. They play an opposite role in determining the time required for ice formation and contribute to the large inter-annual variability observed in ice phenology. More intense cooling, indeed, accelerates the rate of decrease of lake surface water temperature (LSWT), while stronger wind deepens the surface layer, increasing the heat capacity, and thus reduces the rate of decrease of LSWT. A statistical characterization of the process is obtained with a Monte Carlo simulation considering random sequences of the energy fluxes. The results, interpreted through an approximate analytical solution of the minimal model, elucidate the general tendency of the system, suggesting a power-law dependence of the pre-freezing duration on the energy fluxes. This simple, yet physically based model is characterized by a single calibration parameter, the efficiency of the wind energy transfer to the change of potential energy in the lake. Thus, SELF can be used as a prognostic tool for the phenology of lake freezing.

## 1 Introduction

Lake ice phenology is listed as an essential climate product by the global climate observing system. Long-term trends in lake ice phenology are indeed robust archives for climate changes and delays in the calendar dates of the freezing process and earlier thawing are well documented (Livingstone, 1997; Magnuson, 2000; Livingstone et al., 2010; Leppäranta, 2015). While long-term trends regarding the decrease in ice duration are clear, ice phenology time series are also characterized by strong interannual variability (Magnuson, 2000) making any short-term prediction of the ice duration challenging.

The freezing time depends on the amount of heat that was stored in the lake during the summertime and the following rate of heat extraction in fall and winter. Both competing processes are driven by atmospheric forcing. If we exclude very





deep lakes, where thermobaric instabilities can increase the complexity of the process, the different phases can be described
as follows. First, the combination of atmospheric cooling and mechanical wind energy extracts the heat stored in the warm
stratified surface layer and progressively deepens the surface layer to the lake bottom, until the lake reaches homothermal
conditions. Neglecting salinity and pressure effects on density, the homothermal condition is necessarily satisfied when the
lake surface water temperature, $T_s$, is equal to the temperature of maximal density, here set at $T_{md} = 4°C$. The dynamics of
the pre-freezing period, here defined as the time when $0 < T_s < T_{md}$, change compared to the preceding period. Indeed, in
the pre-freezing period, the timing of ice formation is driven by a competition between stabilizing cooling processes (negative
heat flux resulting from seasonal decline in solar radiation, which can be correlated also to air temperature, $T_a$, being colder
than lake temperature) and destabilizing processes (mainly wind). The exact freezing time, occurring when a thin layer at
the surface reaches $0°C$ with a stable temperature gradient below, is dominated by the contribution of the air temperature in
the non-penetrative heat flux. However, both penetrative radiation and wind stress can balance the non-penetrative heat flux
by mixing the previously stratified surface layer, thereby delaying ice formation. Said differently, the interaction between the
different forcing terms (i.e., wind stress and air temperature) will determine the amount of heat to be extracted before the lake
begins to freeze.

The modern approach to predict freezing time consists in using one-dimensional (1D) hydrodynamic models (Liston and
Hall, 1995; Duguay et al., 2003; Dibike et al., 2011; MacKay et al., 2017; Hipsey et al., 2019; Gaudard et al., 2019) coupled
with an ice module (Leppäranta 1993). Such models can be used in prognostic mode, but require a large amount of information
to be measured near the lake to resolve the heat budget. Hence, it remains challenging to accurately estimate ice phenology in
lakes at global scale based on such deterministic models.

The alternative statistical approach, historically initiated in the first part of the XX century, consists in simplifying the
problem by assuming that air temperature is the main driver for ice formation. For instance, ice formation was predicted by
an integration of negative degree days by Bilello (1964). This approach was further extended by Franssen and Scherrer (2008)
with the addition of the mean lake depth as a secondary explanatory variable for ice formation in Swiss lakes. Yet, uncertainties
related to this integral approach have been already discussed by Rodhe (1952), who developed a relationship between weighted
air temperatures and ice formation over the cooling period. A similar approach was then proposed by Leppäranta (1993) with
an analytical slab model based on the linearization of the heat flux. This slab model, as well as other statistical methods
(Weyhenmeyer et al., 2011), takes into account the temporal evolution of the air temperature, $T_a$. Yet, all those models are
intrinsically based on the assumption that the interannual variability of other parameters contributing to the heat budget remain
small compared to the change in $T_a$, or covary with $T_a$. The temporal competition between stabilizing and destabilizing factors
in the pre-freezing phase are thereby not explicitly accounted for in most statistical models.

In this study, we develop a minimal model to predict the time of ice formation based on time series of meteorological
variables. It is important to note that we focus only on the pre-freezing period, i.e., from the day in which the lake is completely
mixed (homothermal) to the day when the surface temperature drops to $0°C$. First, we test the model against the results of a
1D numerical model, Simstrat (Goudsmit et al., 2002; Gaudard et al., 2019), calibrated with in-situ observations for five Swiss
lakes, and then we exploit the simple structure of the minimal model to infer some statistical properties of the freezing process.





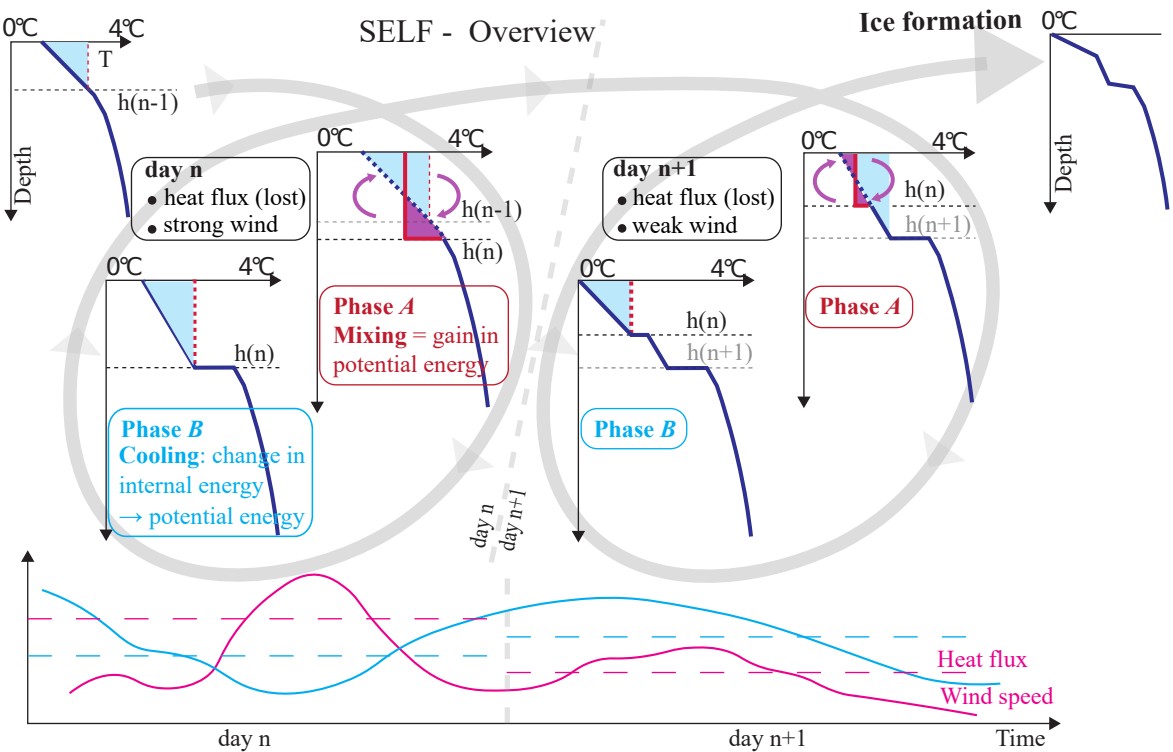

**Figure 1.** Conceptual sketch of the minimal model describing the main processes in the pre-freezing period. Starting from stratified initial conditions ($T$ profile on the top left) at day $n$, part of the wind energy is used to mix the surface layer (phase A). This step sets the thickness of the surface layer, which stratifies due to the heat loss (phase B). The two phases are repeated for day $n + 1$, until the wind stress becomes low enough (reducing the thickness of the surface layer) and the cooling strong enough, so that the temperature at the surface may drop below 0°C, thus forming an ice sheet. The resulting $T$ profile (on the right) tends to become curved (with a concave shape) because, on average, the surface layer becomes thinner when the stratification is stronger.

## 2 Formulation of the minimal model

### 2.1 Phenomenological description

The minimal model (Stratification Energy before Lake Freezing – SELF) simulates the two main processes affecting the development of inverse stratification in the pre-freezing period: the loss of thermal energy due to atmospheric cooling and the input of mechanical energy due to wind stress on the lake surface. The separation of the two processes over a relevant time scale is the core of the minimal model. First, we describe it qualitatively, and then we formulate the mathematical model in the following section. Further details about the simplification of the energy balance are discussed in the Supplementary Material.

The evolution of the stratification modeled by SELF is illustrated in Figure 1. Starting from a stratified water column, the wind stress provides an amount of mechanical energy that mixes the surface layer (phase A), making the temperature uniform





and conserving the thermal energy. The layer's thickness $h$ is determined by balancing the change of potential energy and the fraction of the mechanical energy that is effectively transferred by the wind during a suitably chosen time scale $\Delta t$. The second step (phase B) describes the variation of water temperature distribution due to the net heat flux: we assume that the surface

layer stratifies following an approximately linear profile of temperature along $h$. When heat is lost, the surface layer stably stratifies because the cooling progressively diffuses downwards. In case of warming below 4°C, instead, the stratification is unstable and is readily mixed in the subsequent phase A. The cycle iterates until ice forms at the surface, typically for weak wind conditions when the surface temperature is cold enough. Note that the sequence of mixing and cooling phases, with the surface layer thickness progressively decreasing, gradually builds up a temperature profile with an approximately parabolic

shape, as will be shown later.

A natural choice for the time step of the proposed model is to consider the energy fluxes integrated over a daily cycle ($\Delta t = 86400$ s, 1 day). This retains the net effect of the heat fluxes that are characterized by the diel periodicity given by the solar radiation input. On the other hand, this also means that the destabilizing effect of surface warming during the warmest hours of the day is not explicitly considered in this model.

The net heat flux exchanged through the lake free surface is computed as the sum of several components [W m$^{-2}$]:

$$H_{net} = H_s + H_a + H_w + H_c + H_e \,, \tag{1}$$

where the terms on the right hand side are, respectively, the downward shortwave radiation, the downward longwave (infrared) radiation (mostly depending on air temperature, $T_a$, and cloud cover), the longwave radiation emitted from the lake (depending only on LSWT, $T_s$), the sensible (convection) heat flux (depending on the difference between $T_s$ and $T_a$, through an exchange

coefficient that is a function of $W$), and the latent (evaporation, condensation) heat flux (eventually depending on $T_s$, $T_a$, and $W$). All these terms are evaluated using the same empirical relations implemented in Simstrat (Goudsmit et al., 2002).

In order to keep the model simple, in the computation of the net heat flux (1) we consider the whole shortwave radiation input to the lake, without distinguishing the fraction that is actually absorbed from the surface layer of thickness $h$ from the fraction that penetrates deeper. This assumption might be inaccurate when the surface layer is shallower than the inverse of the

extinction coefficient.

## 2.2 Mathematical formulation

In this section we formulate the model, which we test against observations and numerical results in the next section. We consider a water column of unit area with variable density $\rho(z,t)$ [kg m$^{-3}$] and temperature $T(z,t)$ [°C], linked via a non-linear equation of state, in a gravitational field with acceleration $g$ [m s$^{-2}$], where $z$ [m] is the vertical coordinate pointing

downwards. The minimal model computes: (i) over which depth the previously stratified water column will be mixed (phase A), and (ii) the final temperature profile in the newly created mixed layer (phase B). The equation of state was simplified neglecting the effect of pressure and salinity, hence $\rho(z) = \rho(T(z))$, and calibrating the coefficients of a parabolic function of $T$ between 0 and 4°C:

$$\rho = a_0 + a_1 T + a_2 T^2 \,, \tag{2}$$





where the coefficients $a_0 = 999.8683$ kg m$^{-3}$, $a_1 = 0.0662498$ kg m$^{-3}$ °C$^{-1}$, and $a_2 = -0.00830968$ kg m$^{-3}$ °C$^{-2}$ were obtained from a quadratic regression of a widely used relation (Martin and McCutcheon, 1999; Read et al., 2011) with root mean square error $= 2.7 \times 10^{-4}$ kg m$^{-3}$, bias $= -1.1 \times 10^{-6}$ kg m$^{-3}$, and respecting that $T_{md} = 3.986$°C.

If not specified differently, we assume that the volume of the lake is conserved even when the density changes. Although this assumption is physically wrong (mass is conserved, not volume), it is routinely adopted in all practically used numerical models and does not significantly affect the final results (for a deeper discussion, see the Supplementary Material).

We start analyzing the processes in phase A. At a given time $t$, the potential energy is computed from the free surface ($z = 0$) to a generic depth $Z$:

$$E_p(Z,t) = -\int_0^Z \rho(z,t)gz\,dz\,. \tag{3}$$

In a well-mixed surface layer of thickness $h$, with uniform temperature $T_m = h^{-1}\int_0^h T\,dz$ and density $\rho_m = \rho(T_m)$, the potential energy is $E_{p,m} = -\rho_m g h^2/2$. Hence, the change of potential energy from a stratified condition to a well-mixed layer (for the same depth $h$) is $\Delta E_p(h) = E_p(h, t+\Delta t) - E_p(h,t) = E_{p,m} - E_p(h,t) > 0$. The demonstration of the latter inequality is given in the Supplementary Material.

The energy required to mix the layer down to a depth $h$ comes from the wind force acting on the lake surface. However, only a small fraction of the wind energy is actually transferred into the change of potential energy $\Delta E_p(h)$, and most of it is eventually dissipated. The estimation of the effective wind energy can be split into two processes: (i) the energy transferred from the wind to the surface currents, and (ii) the transfer of the kinetic energy into the change of potential energy of the water column.

Concerning the first process, the wind power [W m$^{-2}$] is usually estimated as

$$P_w = \tau W = \rho_a C_D W^3\,, \tag{4}$$

where $\tau$ [N m$^{-2}$] is the wind shear stress on the lake surface, $W$ [m s$^{-1}$] is the wind speed, $\rho_a$ is the air density, and $C_D$ [-] is the wind-dependent drag coefficient. By integrating the wind power during a day, we obtain the wind energy $E_w = \int_{\Delta t} P_w\,dt$ [J m$^{-2}$ day$^{-2}$]. A fraction of this energy is transferred to the lake in terms of mechanical work $E_k = \int_{\Delta t} \tau U\,dt$ [J m$^{-2}$ day$^{-1}$] that increases the kinetic energy of the wind-driven currents at the lake surface, where $U$ [m s$^{-1}$] is the surface water velocity (precisely, its component in the direction of the wind). Here, we introduce a first efficiency factor as $E_k = \eta_1 E_w$. A preliminary estimate of this ratio, based on the dependence of $U$ on $W$, is provided in the Supplementary Material. We note that the definition of $E_w$ is not unequivocal since it depends on the height where the wind speed is measured, but has the advantage of being simple; conversely, the definition of $E_k$ is rigorous but the velocity $U$ is more difficult to estimate properly.

Then, we focus on the second process, that is, the transfer of the kinetic energy into the change of potential energy of the water column. Only a fraction of the whole kinetic energy $E_k$ is transformed into potential energy of the water column (Kullenberg, 1976). A large fraction is dissipated due to internal friction (turbulence and eventually viscous dissipation at the small scales), and another fraction is used to accelerate the flow in the well-mixed layer (possibly considering also the




entrainment of calm water if the layer becomes deeper). The remaining effect is quantified through a second efficiency as $\Delta E_p = \eta_2 E_k$. All basin-scale dynamic phenomena (up- and downwelling, seiches, and so on) eventually contribute to this term.

It is complicated to provide an independent quantification of the two coefficients $\eta_1$ and $\eta_2$ exactly. Instead, we refer to a single calibration parameter in the form of the global efficiency $\eta$ of the energy transfer from the wind to the change of potential energy in the lake, such that:

$$\Delta E_p(h) = \eta E_w \,, \tag{5}$$

where $\eta = \eta_1 \eta_2$. Thus, given the wind energy, it is possible to compute the depth $h$ of the surface layer that is mixed due to the
wind action.

The formation of the stratification (phase B) is also difficult to characterize in simple terms because it depends on how the temperature changes with depth: the vertical (turbulent in many cases) diffusion of heat interacts with the penetration of short wave radiation and the convective flux. In our simplified model, as a first approximation, we assume that a linear temperature profile develops in the well-mixed layer $h$, with the temperature unchanged at the depth $h$ and the largest variability at the
surface. The net heat flux across the free surface (assumed positive for cooling, when the flux is directed from the lake to the atmosphere) includes the incoming shortwave radiation and the other heat fluxes exchanged with the atmosphere. The energy per unit area $E_c$ exchanged during the interval $\Delta t$ [J m$^{-2}$ day$^{-1}$] is computed by integrating the net heat flux in time, $E_c = \int_{\Delta t} H_{net} dt$.

Given the thickness $h$ of the surface layer and the heat loss $E_c$, it is possible to compute the difference between the undis-
turbed temperature at the bottom of the layer, $T|_{z=h}$, and that at the surface, $T|_{z=0} = T_s$,

$$\Delta T = T|_{z=h} - T|_{z=0} = \frac{2E_c}{\rho_0 c_p h} \,, \tag{6}$$

where $\rho_0$ is a reference value for water density. Hence, the change in the $T$ profile modifies the potential energy of the system (phase B in Figure 1) leading to a condition where an input of external energy (wind) is required to mix it again.

## 2.3   Cumulated energy and duration of the pre-freezing period

Having presented the minimal model, we focus then on the expected output. We define as $n_d$ the number of days between the start of the simulation (on the day when $T_s$ permanently falls below $T_{md}$) and the first time when $T_s < 0°$C, i.e. the duration of the inverse stratification (pre-freezing) period. Referring to this period, we define the cumulative values [J m$^{-2}$] of the mechanical and thermal energy

$$\Delta E_p^{(n_d)} = \sum_{i=1}^{n_d} \Delta E_p(\text{day } i) \,, \tag{7}$$


$$E_c^{(n_d)} = \sum_{i=1}^{n_d} E_c(\text{day } i) \,. \tag{8}$$

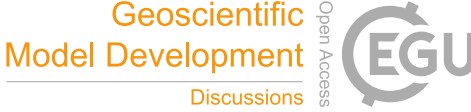

Moreover, we are going to relate our results with other approaches based on negative degree days, $D$ [°C days], defined as follows

$$D = -\sum_{i=1}^{n_d} \min\{0, T_a(\text{day } i)\},\tag{9}$$

where $T_a$ is the daily averaged air temperature expressed in °C. The summation is done only for values of $T_a < 0$°C (Franssen and Scherrer, 2008).

### 2.4 Approximate explicit solution

The model SELF does not admit an analytical solution in closed formed. A simplified explicit dependence between $\Delta E_p^{(n_d)}$ and $E_c^{(n_d)}$ can be obtained only by introducing other additional assumptions that are not fully realistic. Bearing in mind that

the obtained solution does not aim at describing real conditions but instead to explore the relative contribution of heat loss and wind intensity, we assume that the daily energy fluxes are constant (hence neglecting the history of the system) and that the density depends linearly on $T$ (not representative of what occurs in the range $0-4$°C, as already noted). Referring the reader to the derivation provided in the Supplementary Material for the details, an approximate quadratic dependence between the cumulated energies is obtained:

$$\Delta E_p^{(n_d)} = k \left( E_c^{(n_d)} \right)^2,\tag{10}$$

with the proportionality coefficient $k \sim O(10^{-15})$ s$^2$ kg$^{-1}$. The very small value of the coefficient $k$ is due to the several orders of magnitude of difference between the heat loss, $E_c^{(n_d)}$, and the mixing energy, $\Delta E_p^{(n_d)}$, amplified by the quadratic dependence.

The approximate analytical solution also provides a direct estimate of the duration of the pre-freezing period,

$$n_d = k^{-1} \langle \Delta E_p \rangle \langle E_c \rangle^{-2},\tag{11}$$

as a function of the averaged daily values of the energy (indicated by angle brackets). Although the unrealistic simplifications introduced to derive such a result, equations (10) and (11) provide a way to interpret the relationship between the strengths of cooling and wind mixing. This solution corresponds to an extension of the well-established estimate of ice freezing probability based on negative degree day. Specifically, the term expressing heat loss is an analog to the negative degree day, while the

addition of a term expressing the mixing energy now includes the delaying effect of wind intensity.

## 3 Methods

### 3.1 Observations

Five Swiss lakes were selected as case studies: Sils, Silvaplana, St. Moritz, Sihl and Joux. The main relevant geographical and meteorological characteristics are provided in Table 1. Those five lakes cover a wide range of different forcing conditions in



**Table 1.** Main geographical and meteorological characteristics of the studied lakes.

| Parameter | Sils | Silvaplana | St. Moritz | Joux | Sihl |
|---|---|---|---|---|---|
| Position [latitude | 46°25'0"N | 46°26'55"N | 46°29'52"N | 46°38'16"N | 47°7'1"N |
| and longitude] | 9°43'51"E | 9°47'38"E | 9°50'18"E | 6°17'4"E | 8°47'0"E |
| Altitude [m asl] | 1797 | 1791 | 1768 | 1004 | 889 |
| Volume [$10^6$ m$^3$] | 137 | 140 | 20 | 145 | 96 |
| Surface area [km$^2$] | 4.1 | 2.7 | 0.78 | 8.53 | 11.3 |
| Max depth [m] | 71 | 77.5 | 44 | 32 | 22 |
| Mean depth [m] | 33.5 | 52 | 26 | 17 | 8.5 |
| Max width [km] | 1.1 | 1.4 | 0.6 | 1.3 | 2.5 |
| Max length [km] | 5.0 | 3.1 | 1.6 | 9.0 | 8.5 |
| Meteorological station | Segl-Maria | Segl-Maria | Samedan | Les Charbonnières | Einsiedeln |
| | (SIA) | (SIA) | (SAM) | (CHB) | (EIN) |
| Number of years | 5 | 5 | 38 | 10 | 7 |
| *Averaged values in the pre-freezing period* | | | | | |
| Duration [days] | 24.4 | 22.4 | 6.2 | 17.2 | 15.4 |
| Air temperature [°C] | -5.7 | -5.1 | -8.4 | -1.8 | -1.0 |
| Wind speed [m s$^{-1}$] | 3.1 | 3.0 | 1.3 | 2.3 | 1.5 |
| Air pressure [mbar] | 815 | 816 | 818 | 897 | 910 |
| Vapor pressure [mbar] | 2.92 | 2.97 | 2.80 | 4.90 | 5.09 |
| Shortwave radiation [W m$^{-2}$] | 56.1 | 54.9 | 61.2 | 37.0 | 44.5 |
| Cloud cover [-] | 0.22 | 0.22 | 0.41 | 0.25 | 0.52 |

the pre-freezing period, varying from mild to cold air temperature and weak to moderate wind intensities (see also the analysis in the Results section). For each lake, wind speed, air temperature, incoming solar radiation, vapor pressure and cloud cover data were taken from the closest meteorological station within the automatic monitoring network of MeteoSwiss (see Table 1).

Lake temperatures are continuously recorded at different depths. For Lake Joux, the mooring consists of 9 temperature loggers (accuracy 0.1°C) equally spaced from 2 m below the surface to the lake bottom; the monitoring system has been in

place since 2013. For Lakes Sils, Silvaplana, St. Moritz, and Sihl, the moorings consist of 11 temperature loggers (accuracy 0.1°C). In the first year (2016), the mooring was designed to follow the evolution of the temperature in the surface layer with the first temperature logger ~5 cm below the surface. The distance to the next sensor was set to be the double of the distance just above. For safety and practical issues, the mooring stopped at a sub-surface buoy 2 m below the surface in the following years.

These datasets provide the necessary information to validate a 1D hydrodynamic model for standard applications related to the evolution of the thermal structure. However, in this case where we aim at considering the LSWT, the distance of the logger closest to the surface is not sufficient to obtain information about the correct timing of ice formation and, hence, to robustly





validate our minimal model. For this reason, a traditional physically based model (which provides the water temperature right at the lake surface) was used as the prototype to compare with.

## 3.2 One-dimensional full model

We used a 1D vertical hydrodynamic model, Simstrat v2.1 (Gaudard et al., 2019), to provide a vertically resolved time series of water temperatures for testing the proposed minimal model. For details about the model structure, we refer the reader to Goudsmit et al. (2002). Here it suffices to mention that the heat fluxes are calculated in the same way as for the minimal model (equation 1); note that this method does not take the atmospheric stability into account. Similarly, wind energy transferred to the lake is estimated with the same wind drag coefficient (Wüest and Lorke, 2003) used in SELF.

Simstrat has already been successfully applied to the five investigated lakes for yearly monitoring of the thermal structure (Gaudard et al., 2019). Here, we specifically calibrated Simstrat to the pre-freezing period. For each lake, the calibration parameters were adjusted based on the first year of observations and the model was validated with the following pre-freezing periods (more details about the performances in the Supplementary Material). The beginning of the pre-freezing period was defined at the time the upper temperature logger reached 4°C.

We acknowledge that even the one-dimensional approach of the mechanistic model cannot accurately reproduce the exact timing of ice formation given the horizontal variability of the ice formation process at the lake surface, typically starting from the shore and propagating offshore over a couple of days (Leppäranta, 2015). Nevertheless, in the absence of detailed information about the spatial distribution of ice in the majority of lakes, 1D models often represent the only deterministic approach consistent with the knowledge available for the investigated system. In this respect, SELF contains an even more simplified description of the vertical stratification process with regard to classical physically based models such as Simstrat.

## 3.3 Calibration of the minimal model

In order to calibrate the wind-to-potential-energy efficiency $\eta$ in the minimal model, we compared the results of SELF with those obtained with Simstrat. Two aspects were considered: the duration of the pre-freezing period, $n_d$, and the difference in daily LSWT, $T_s$, during this period. We weighted the two factors to define the error to be minimized:

$$err = \left| n_d^{\text{SELF}} - n_d^{\text{Simstrat}} \right| + \omega_T \sqrt{\frac{1}{n_{d,\min}} \sum_{i=1}^{n_{d,\min}} \left( T_{s,i}^{\text{SELF}} - T_{s,i}^{\text{Simstrat}} \right)^2}, \tag{12}$$

where $n_{d,\min} = \min \left\{ n_d^{\text{SELF}}, n_d^{\text{Simstrat}} \right\}$ and $\omega_T = 1 \ °C^{-1}$ is the (arbitrarily chosen) relative weight of the temperature deviation during the simulation period with respect to the freezing time difference. The optimal value of the parameter $\eta$ was obtained by minimizing $err$ for each lake using a bifurcation algorithm.



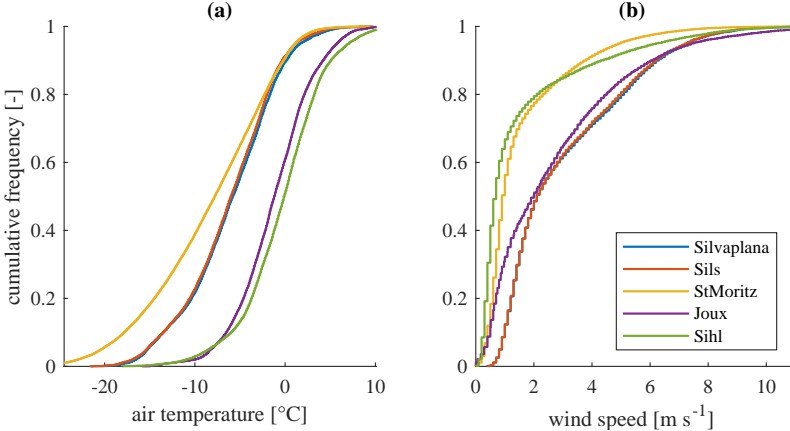

**Figure 2.** Comparison of the meteorological conditions among the five lakes: cumulative distribution of air temperature (a) and wind speed (b) in the extended (i.e., considering 15 days after the latest ice-on date for each lake) pre-freezing period, computed using the whole series of available measurements (see Table 1). The distributions are plotted between the values 0.01 and 0.99 of cumulative frequency. Note that the data of Lakes Silvaplana and Sils are almost coincident because they refer to the same meteorological station and the only difference lies in the slightly different duration of the pre-freezing period in the two lakes.

## 4 Results

### 4.1 Climatological characterization of the pre-freezing period

In this section, we present the statistics of the wind speed and air temperature, considered as the main meteorological drivers of lake freezing, for the five selected Swiss lakes. Our analysis focuses on the pre-freezing period, which we extend for each lake from the day of homothermal conditions (in each year, the latest date when LSWT drops below 4°C) to 15 days after the latest date of ice-cover formation in the available results (hereafter, this period will be qualified as "extended"). The cumulative distributions of air temperature and wind speed shown in Figure 2 indicate a wide range of forcing conditions, although the five lakes lie in similar geographical region.

The two lakes that are located around approximately 1000 m a.s.l. (above sea level), Lake Sihl and Lake Joux, have a median air temperature of -1.0°C and -1.8°C over the extended pre-freezing period, respectively. Air temperature in the higher altitude lakes (∼1700 m asl) is almost constantly below zero in the same period, with median air temperature of -8.4°C for Lake St. Moritz, and -5.7 and -5.1°C for Lakes Silvaplana and Sils (Figure 2a). Note that for the last two lakes there is only one meteorological station, and the median air temperature depends on the extended pre-freezing averaging window, which differs between lakes. From these results, we expect $n_d$ to be longer for Lakes Joux and Sihl, and likely the shortest for Lake St. Moritz. Wind intensity also varies over the investigated system, with median wind speed and wind power being respectively two and eight (wind power depending on the third power of wind speed) times stronger over Lakes Joux, Silvaplana and Sils than over Lakes St. Moritz and Sihl (Figure 2b). When adding wind information, we now expect Lake Sihl to have a shorter





$n_d$ than Lake de Joux, and similarly Lake St Moritz having the shortest $n_d$ over the Upper Engadin lakes. The large range in the observed forcing allows for future global application of our regional process-based study.

### 4.2 Performances of Simstrat

We compare the temperatures simulated with Simstrat and the different near-surface temperature loggers during the pre-freezing period (Figure 3a). The model performances are summarized with an $R^2 = 0.88$ and RMSE $= 0.19$°C. From Figure 3a, where the data become very sparse for LSWT close to 0°C because the logger is not at the surface, we immediately see that we lack information near the surface, which is needed to calibrate SELF in a proper way. In 2016, the loggers installed near the surface in Lake Sihl could measure the temporal evolution of this layer down to a temperature of 0.5°C (Figure 3b). The

simulated temperatures with Simstrat follows the general trend with a RMSE $= 0.23$°C. Interestingly, the change in slope at the end of the period is correctly reproduced by the model. This evidence further supports the use of Simstrat to simulate the evolution of the thermal structure during the pre-freezing period.

In order to improve the agreement between the predicted evolution of the thermal structure and the observations during the pre-freezing time, the deterministic model would require more accurate meteorological data from stations located within

the lake or close to the shores, which is not the case in general. An optimization of the initial conditions would have also improved the model, but we opted to start with homothermal conditions. Nevertheless, given the unavoidable uncertainties in the determination of the forcing energy fluxes and their relationship with the response of the lakes in the actual cases, we decided to rely on the results of the Simstrat model completely, and to use those outputs as the reference case to compare with.

### 4.3 Evolution of the stratification in the minimal model

The shape of the temperature profile in conditions of inverse stratification is often characterized by a curved, approximately parabolic profile. Interestingly, such a shape is correctly reproduced by SELF because of the sequence of wind-driven mixing and cooling-induced partial stratification, as shown in the example for Lake Sils in Figure 4. In fact, in the initial phase the mechanical energy provided by the wind is sufficient to mix the water column deep down. However, as the water column becomes more stratified during a sequence of cold days, the layer that can be mixed by the wind becomes thinner and thinner.

As a consequence, the process of LSWT cooling accelerates, until reaching 0°C at the surface, and the resulting profile has a curved, concave shape.

The detailed analysis of the plots in Figure 4 (where the profiles simulated with Simstrat are added for a comparison) allows us to understand how SELF works on these selected days. In day 8, a stronger wind thickens the surface layer when the stratification is weak: the change in SELF is discontinuous (as it can be detected comparing the solid line with the dotted line

referring to the previous day), but the depth that is affected coincides with the end of the stratification in Simstrat. In calm conditions (day 15), the linear profile assumed by SELF in the surface mixed layer well approximates the continuous Simstrat profile. A strong wind event on day 19 produces a clear mixed layer of the same depth in the two models. Finally, the lake freezes on day 21 with an overall profile characterized by a similarly shaped profile.



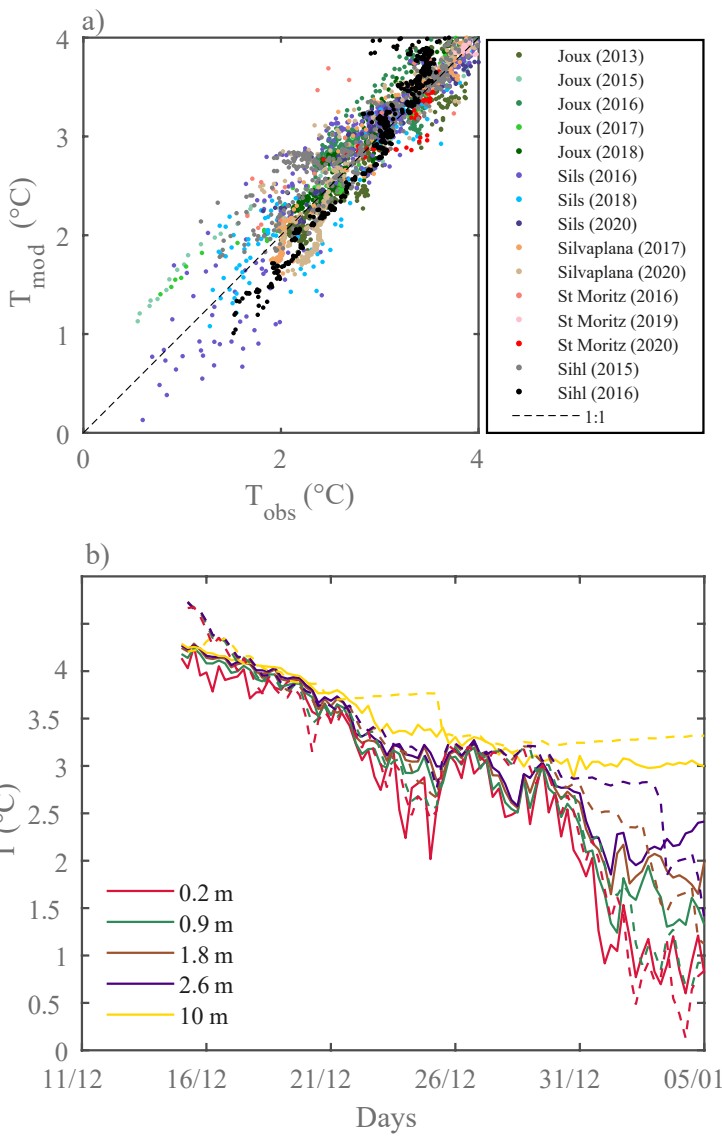

**Figure 3.** Performances of Simstrat in reproducing temperature observations. (a) Scatter plot of the observed and modelled temperature (Simstrat) during the pre-freezing period for the five studied lakes. For this figure, the data are extracted with a period of 8 h. (b) Temporal evolution of the temperature in the first 10 m as observed by the thermistors (continuous line) and by the numerical model Simstrat (dashed line) for Lake Sils.





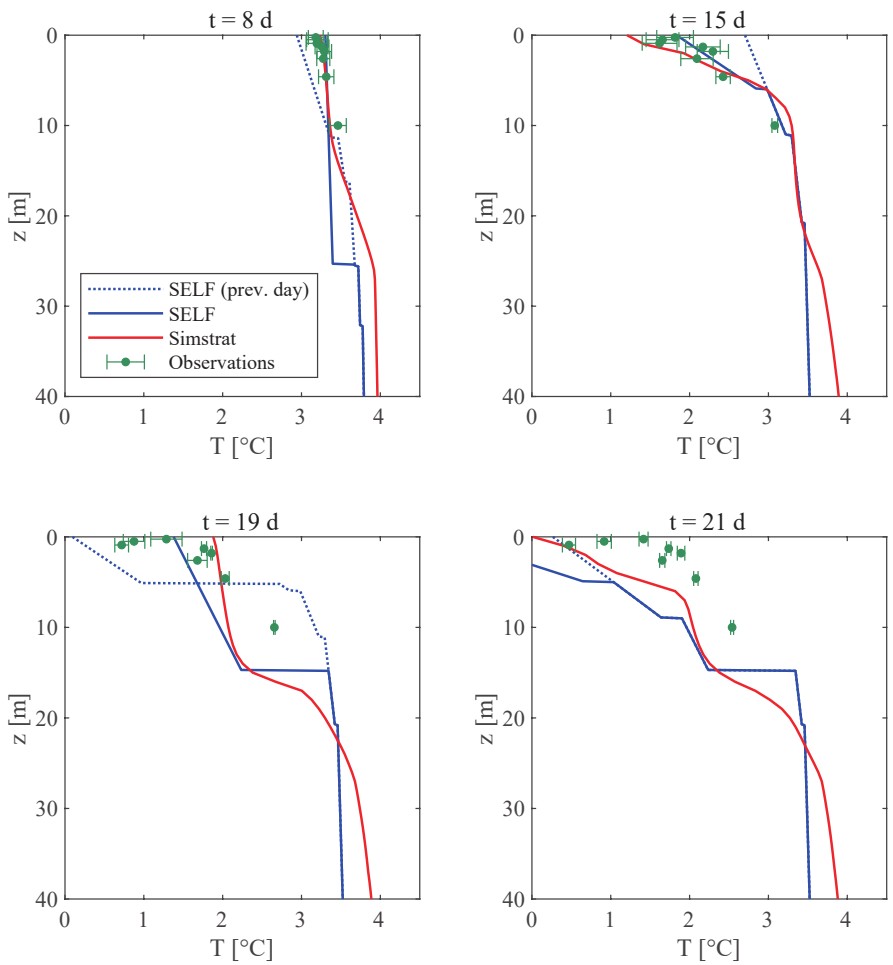

**Figure 4.** Evolution of the temperature profile in the pre-freezing period for Lake Sils between 18 December 2016 and 08 January 2017. The plots represent the daily profiles obtained with SELF compared with the profiles at midnight computed by Simstrat. The dotted lines represent the SELF model's temperature profile on the previous day. The symbols represent the available observations, with the error band equal to one standard deviation of the temperature during the day.





For the selected year there are also in-situ observations available: they are included in the different subplots of Figure 4
making it possible to quantitatively evaluate the performance of the two models. There is an excellent agreement until day 18
(the whole sequence is provided in the Supplementary Material). On day 18, both models responded to the increase in wind
intensity by mixing the surface layer down to 15 m. This wind-mixing event is not observed in the in-situ lake temperatures
data. Indeed, the temperature profile remains stratified during this period. The deviation from the observations of both models
forced by the same atmospheric dataset shall not be interpreted as a deficiency of the models, but rather as a need to provide
more accurate on-lake meteorological data.

### 4.4 Performances of the minimal model

The duration of the pre-freezing period $n_d$ was estimated in three different ways. In all cases, the wind energy was externally
prescribed, while differences exist in the computation of the heat loss, which depends on the LSWT, i.e. on the result of
the model itself. Thus, three options for the quantification of the heat loss were selected: (1) using the LSWT computed by
Simstrat (case "Ts-Simstrat"), hence having an externally prescribed heat loss; (2) using the LSWT computed by SELF (case
"Ts-SELF"), thus using the minimal model as a prognostic tool; (3) using the LSWT from SELF as in the previous case, but
forcing the model with constant values of the meteorological variables averaged over the extended pre-freezing period (case
"Ts-SELFav").

The results are shown in Figure 5, which reports the parity diagrams comparing Simstrat (assumed as the truth in this case)
and SELF, with the three options for the computation of the heat loss. The overall agreement is very good for the "Ts-Simstrat"
case (Figure 5a, Pearson's $r^2 = 0.95$), and even better for the "Ts-SELF" case (Figure 5b, $r^2 = 0.97$), with a decay of the
performances, as expected, for the averaged "Ts-SELFav" case (Figure 5c, $r^2 = 0.76$). The SELF model provides realistic
values of $n_d$ over a broad range of pre-freezing periods extending from 2 to almost 40 days for the five investigated lakes, a
performance that is especially surprising considering the simplicity of the minimal model. The improvement in the "Ts-SELF"
case can be likely ascribed to the explicit consideration of the feedback that exists with LSWT in the determination of the heat
fluxes from the lake to the atmosphere.

The values of the wind-to-potential-energy efficiency $\eta$ calibrated for the different lakes are variable (Table 2), with higher
values for shallow lakes (Lakes Joux and Sihl) and lower values for the deepest lakes on the upper Engadine valley (Lakes Sils
and Silvaplana). Lake St. Moritz, although being of intermediate depth, is characterized by a value of $\eta$ that is small, but it is
also the lake with the smallest surface area, which reduces the wind fetch. Yet, we recall that $\eta$ connects the wind energy to
the lake mixing and thereby has not only a physical interpretation for the stratification process leading to ice formation, but
also serves as calibration parameter for the wind forcing. In this respect, wind speed is not measured from a lake buoy, but at
the lake shore or even farther in the case of Lake St. Moritz, where the nearby station is separated from the lake by a hill. The
identification of the various factors affecting the efficiency $\eta$ would require the analysis of a more extended database of lakes.



**Figure 5.** Scatter plot of the duration of the pre-freezing period $n_d$ predicted by the minimal model (SELF) vs. the complete 1D model (Simstrat) using the net heat flux: (a) simulated by Simstrat; (b) reconstructed based on the LSWT estimated by SELF (prognostic mode, i.e., based only on purely meteorological data); (c) reconstructed using averaged meteorological quantities (see Table 1).





**Table 2.** Calibrated parameters of the SELF model for the investigated lakes.

| Parameter | Sils | Silvaplana | St. Moritz | Joux | Sihl |
|---|---|---|---|---|---|
| $\eta$ (Simstrat*) | $2.771\times10^{-4}$ | $2.310\times10^{-4}$ | $0.954\times10^{-4}$ | $0.757\times10^{-4}$ | $0.961\times10^{-4}$ |
| $\eta$ (prognostic*) | $3.137\times10^{-4}$ | $2.479\times10^{-4}$ | $0.974\times10^{-4}$ | $1.091\times10^{-4}$ | $1.341\times10^{-4}$ |
| $\eta$ (averaged*) | $3.780\times10^{-4}$ | $2.712\times10^{-4}$ | $1.788\times10^{-4}$ | $1.235\times10^{-4}$ | $0.717\times10^{-4}$ |
| $k$ [$10^{-15}$ s$^2$ kg$^{-1}$] ** | 0.56 | 0.50 | 0.12 | 0.73 | 1.3 |

* The efficiency $\eta$ of energy transfer of $E_w$ to $\Delta E_p$ was calibrated for the five lakes in three different ways: using net heat fluxes directly from Simstrat, using LSWT from SELF to compute the fluxes (prognostic mode), using seasonally averaged meteorological variables in the prognostic mode.

** The coefficient $k$ for the simplified analytical model (equation 10) was calibrated on the median value of the distribution of random sequences in SELF (further details in the Supplementary Material).

### 4.5 Monte Carlo analysis with SELF

The simplicity of the SELF model allows for the characterization of the pre-freezing period by exploring a large number of simulations following a Monte Carlo approach. We performed 100'000 runs with SELF (in the prognostic mode) for each lake, in which the time series of the wind energy and of the meteorological variables used to quantify the heat loss are randomly sorted from the actual sequences over the whole dataset of the extended pre-freezing period. The goal is to investigate the influence of the wind and of the cooling during the pre-freezing period and eventually to provide a simple analytical solution to predict the duration of the pre-freezing period.

For each run, the freezing time $n_d$ is associated with the cumulated values of heat loss $E_c^{(n_d)}$ and mixing energy $\Delta E_p^{(n_d)}$ (depending on wind energy $E_w$ through the efficiency $\eta$), and represented in a diagram using color-scaled dots (see Figure 6 for Lake Silvaplana, where only 1'000 random runs are plotted for clarity). This visualization illustrates that the length of the pre-freezing period is controlled by the amount of heat extracted, but is also dependent on the input of the wind energy. Figure 6 also shows some examples of trajectories of the random runs (black lines) in the $\Delta E_p^{(n_d)}$-$E_c^{(n_d)}$ plane, and one sequence characterizing an actual winter (red line). The details of this single year are presented in Figure 7, which shows the whole sequence of LSWT values simulated by Simstrat and SELF, respectively, together with the daily averaged net heat flux and wind power. The same analysis has been developed for the other lakes, as well: please refer to the Supplementary Material for the corresponding figures.

Exploiting the total number of the Monte Carlo runs, it is possible to characterize the behavior of the process in a more exhaustive way. Figure 8 shows the results in the $\Delta E_p^{(n_d)}$-$E_c^{(n_d)}$ plane, while Figure 9 in the $\langle \Delta E_p \rangle$-$\langle E_c \rangle$ plane, i.e. the mean daily energy fluxes. The two figures are built in the same way: subplot (a) shows the distribution of all combinations; subplot (b) reports the main result, i.e. the mean duration $n_d$ of the pre-freezing period; subplot (c) the standard deviation of $n_d$; and subplot (d) the latter value normalized with the mean.

The analysis of the distribution of the points in the cloud in Figure 6 suggests a relationship between $\Delta E_p^{(n_d)}$ and $E_c^{(n_d)}$, with $n_d$ growing as the values of the two quantities increase. The duration $n_d$ grows from the lower left to the upper right



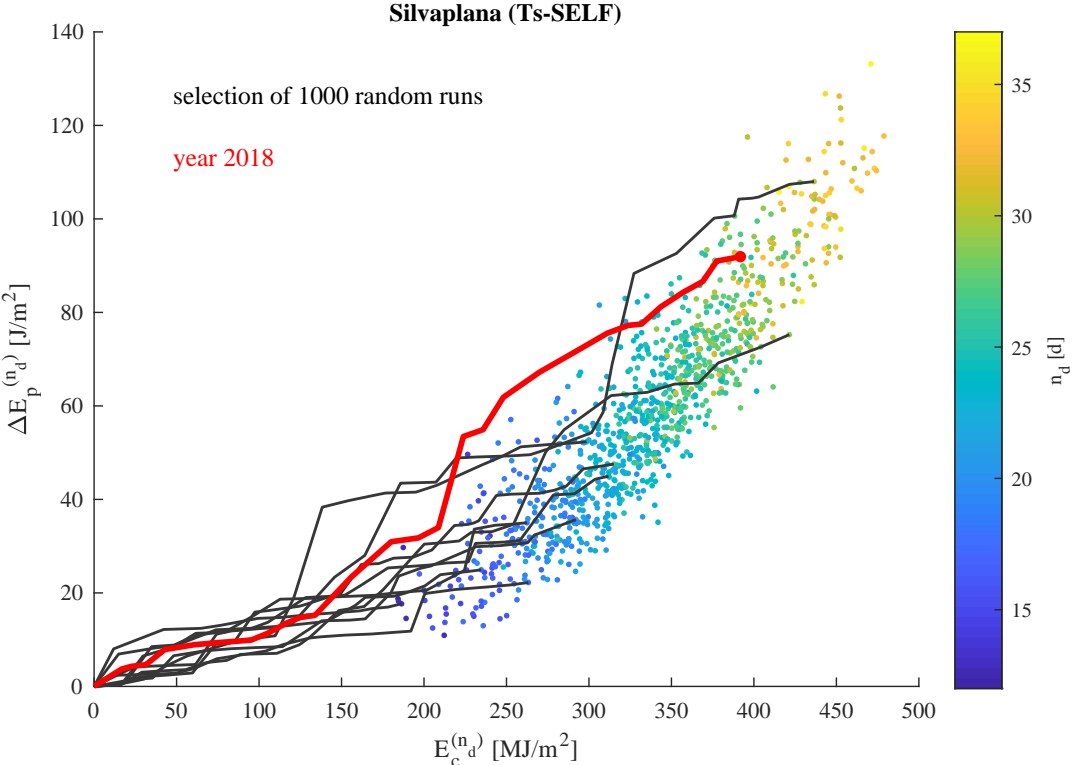

**Figure 6.** Cumulative mixing energy $\Delta E_p^{(n_d)}$ vs. lost energy $E_c^{(n_d)}$ for Lake Silvaplana, as obtained from the random model. The color scale represents the duration of the pre-freezing period. Black trajectories correspond to random sequences; the red trajectory represents the pre-freezing period of a real winter (see Figure 7).

corner of Figure 8b, and the analytical curve from equation (10), shown with a red line, captures well the general tendency of the minimal model's runs. It is possible to identify an upper and a lower boundary, which represent, for a given wind history

(which translates here into the cumulated mixing energy $\Delta E_p^{(n_d)}$, hence along a horizontal line), the minimum and maximum cumulated loss of heat, respectively, under which ice can form. If more heat is extracted daily, the ice will form before, thus moving the point left and down; if less heat is extracted, ice will form later (point moving right and up), or not forming at all if the process takes too long and the spring warming arrives. Figure 8d shows that the results are more variable, in relative terms, for shorter pre-freezing period, for which the actual history of the meteorological forcing matters even more.

The analysis of the results in the plane of the daily values (Figure 9b) shows that the effect of the wind increases as it becomes faster and the cooling weaker (left upper region), where the isolines with constant $n_d$ (represented by black lines) become more vertical. Hence, windy lakes will take longer to freeze, especially if they are not in a very cold climate. The general trend is predicted reasonably by the simplified analytical solution (colored lines with numbers representing $n_d$ values). The standard deviation shows higher values for windy and warm lakes (Figure 9c) because of the stronger influence of the



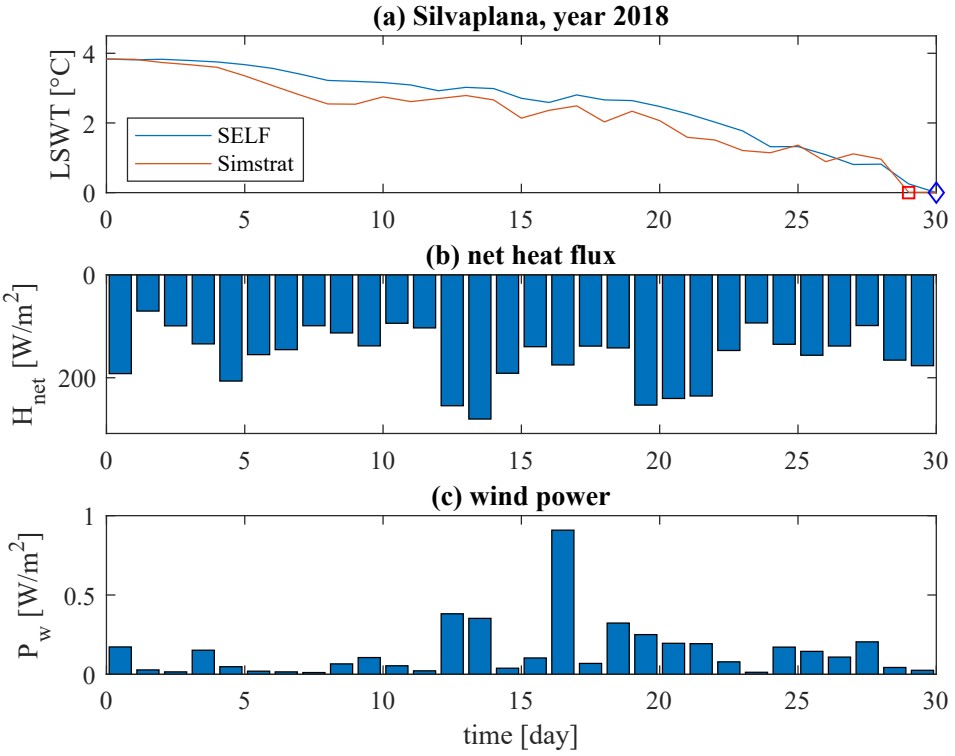

**Figure 7.** Example of the dynamics in the pre-freezing period for winter 2018/19 in Lake Silvaplana corresponding to the red trajectory in Figure 6: (a) comparison of the results of Simstrat and SELF (in the prognostic mode) with symbols representing the ice-on day; (b) daily-averaged net heat flux (positive values for cooling); (c) daily-averaged wind power.

wind history, with moderate variability in terms of its relative value (Figure 9d). The figures for the other lakes are available in the Supplementary Material.

## 5    Discussion

### 5.1    Factors controlling the freezing time

The time series illustrated with a red trajectory in Figure 6 shows that for a given energy loss (here ∼400 MJ m$^{-2}$), a change in mixing energy from ∼60 J m$^{-2}$ to  100 J m$^{-2}$ will delay ice formation by about 5 days (see Figure 8b) starting from the homothermal conditions. In order to have a reference to compare with, the observed trend in the ice-on date is 5.8 days per century in the northern hemisphere over the last 150 years (Magnuson 2000); note that this delay is affected also by the shift of the day when homothermal conditions are realized. A variation of +67% in the cumulated mixing energy (from ∼60 J m$^{-2}$ to ∼100 J m$^{-2}$) corresponds to a change of mean wind speed of approximately +19%, which is a relatively smaller change

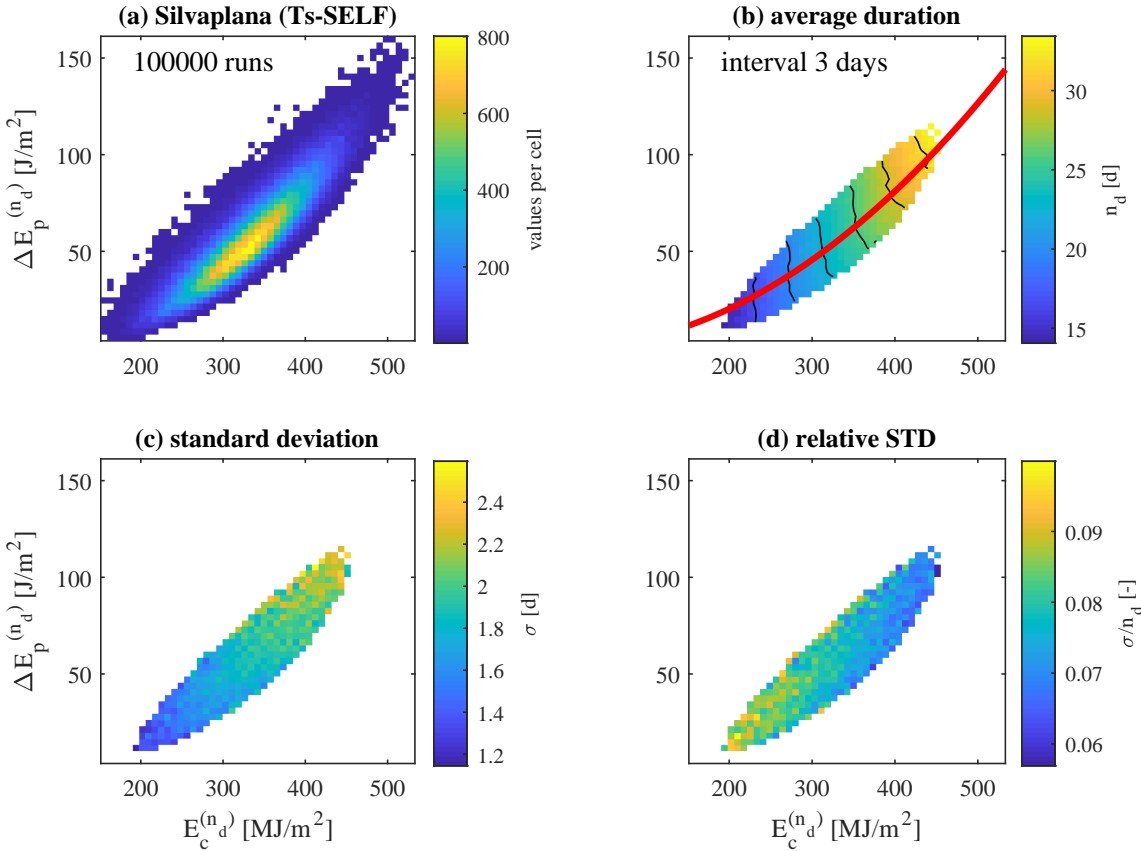

**Figure 8.** Distribution of the results of the SELF model applied to 100'000 random sequences in Lake Silvaplana, as a function of the cumulative values of the energies, $\Delta E_p^{(n_d)}$ and $E_c^{(n_d)}$: (a) number of results per computation cell; (b) average duration in days, $n_d$, of the pre-freezing period, with black lines representing constant $n_d$ with the interval indicated in the plot; (c) standard deviation of $n_d$ in the cell; (d) relative standard deviation. The thick red line in panel (b) represents the theoretical dependence obtained by means of the simplified analytical model (equation 10, using the coefficient $k$ reported in Table 2).



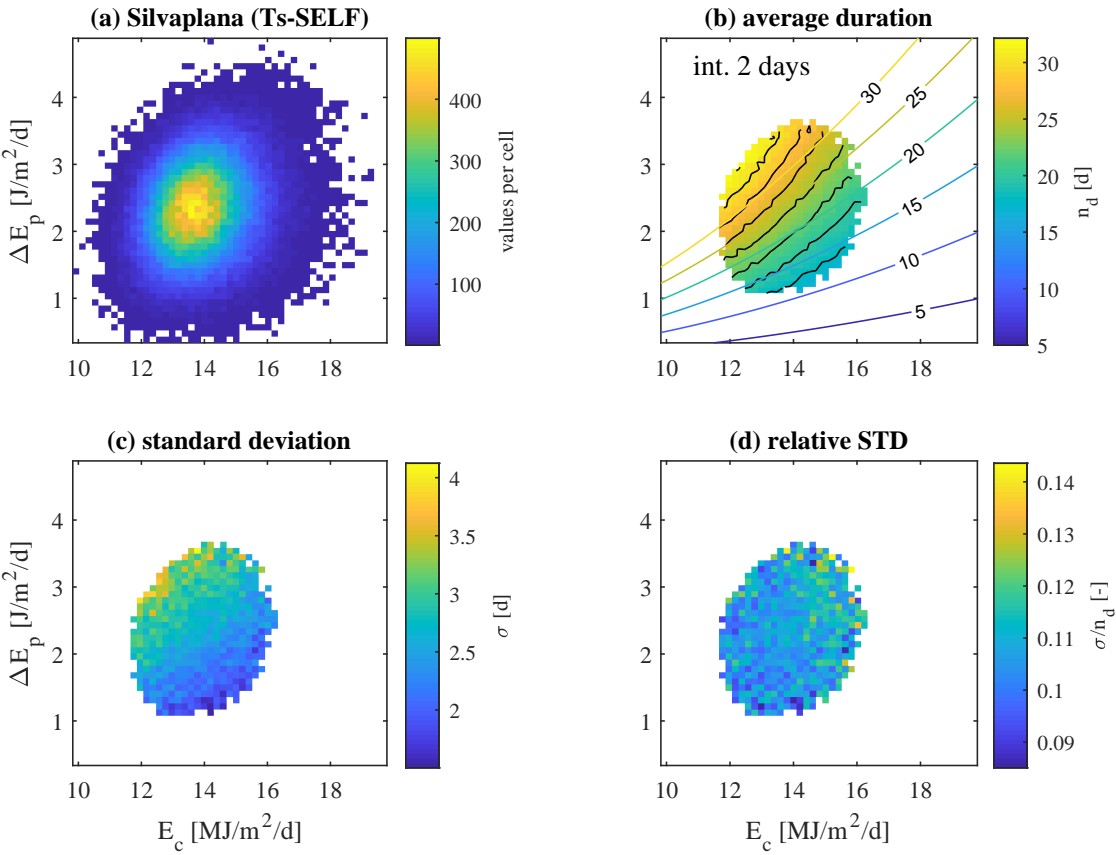

**Figure 9.** The same results as in Figure 8, but in terms of average daily values of the energies, $\langle \Delta E_p \rangle$ and $\langle E_c \rangle$: (a) number of results per computation cell; (b) average duration in days, $n_d$, of the pre-freezing period, with black curves representing constant $n_d$ with the interval indicated in the plot; (c) standard deviation of $n_d$ in the cell; (d) relative standard deviation. The colored lines (colors associated with the color bar) in panel (b) represent the theoretical $n_d$ obtained by means of the simplified analytical model (equation 11; see also Figure 8) with numbers indicating the estimated values of $n_d$.





in the forcing given the cubic dependence. The timing also largely depends on the daily sequence of the wind power and heat exchanges (Figure 7b,c), with single wind peaks producing much larger peaks in the mixing energy (and consequently on its cumulated value), again due to the cubic dependence.

Our results suggest that taking only into account the cooling (using for instance negative degree days, which depend solely on air temperature) may not explain the inter-annual variability in the ice formation, and that the variability due to wind speed

can be as large as the change resulting from a century of increase in air temperature. Adding the competition between cooling processes stratifying the water body and wind momentum destratifying the water body, as in SELF, allows for estimating the timing for ice formation more accurately.

SELF can be used to better understand long time series of ice formation in lakes (Magnuson 2000) and specifically to decouple the inter-annual variability from the long-term climate-change induced trend, which shifts predominantly the day

where homothermal conditions occur. The analysis based on the random sequences suggests that the influence of the wind increases for lower values of the heat loss. Thus, due to the increasing air temperature associated with climate change in many regions on Earth, we expect the inter-annual variability in the timing for ice formation to continuously increase in the future. We incidentally note that this effect might hinder the lake ice phenology to be seen as sentinel for climate changes.

In this study, we could not assess the role of lake depth in controlling the freezing process. This is an outcome from the

investigated lakes, which are all deep, with maximum depth ranging from 22 to 77 m. Yet, water depth may become an important driver when the thickness of the surface mixed layer frequently encompasses the whole water depth.

## 5.2 Comparison and limitations of models to predict the freezing time

SELF is a minimal process-based model for predicting ice formation in lakes. Considering the stratification induced by cooling and the mixing induced by wind in an energy balance is a step forward compared to a more traditional accounting of air temper-

ature through negative degree-days (Franssen and Scherrer, 2008), statistical air temperature models (Livingstone and Adrian, 2009) or regression-tree based prediction (Sharma et al., 2019). Those models have to assume that all the other parameters acting at the air-water interface, such as the wind action, stay constant over time. As a result, those approaches are not able to correctly predict the duration of the pre-freezing period: the correlation of $n_d$ with negative degree days, $D$, is generally poor for these lakes, with the exception of Lake Sils for which it is decent (see the details in the Supplementary Material).

We have reported that the stochasticity in the wind speed and air temperature contributes to the timing of ice formation, and this element cannot be neglected in the majority of applications. Our energy-based model efficiently copes with this issue by comparing, on a daily time scale, the cooling-induced stratification and wind-induced mixing: the chronological sequence of these two factors has to be necessarily taken into account to correctly predict ice formation.

Describing the competition between stratification and destratification processes is typically the strength of 1D hydrodynamic

models. Yet, SELF is computationally simple and thousands of runs can be simulated in a few seconds, which is impossible with classical 1D hydrodynamic models. SELF has however the same limitations of any 1D model: for instance, it does not consider the horizontal variability of ice formation, which typically starts from the shore and propagates offshore, introducing an uncertainty in the definition of a univocal ice-on date. In this respect, SELF was tested in five perialpine lakes of various





sizes, yet sharing similar morphologies. The validity of SELF, and more generally of any 1D model, to very large systems
remains to be demonstrated; another issue may arise from lakes with extended shallow areas. The deviation from the classical
1D framework of the heat budget as a function of morphology and latitude was recently shown for the end of the ice-covered
period when lake dynamic is influenced by radiatively-driven convection (Ulloa et al., 2019; Ramón et al., 2020). In the case of
the pre-freezing period, the amount of heat stored in the sediment in the shallow area may affect the system (Fang and Stefan,
1996). However, the buffer role of the sediments in the heat budget was not investigated here. In this respect, large differences
between SELF and observations of the pre-freezing duration can also be interpreted as interesting signatures of deviation from
the classical 1D energy budget framework with other processes to be specifically investigated.

    A further limitation of the model that might play a role in the proximity of homothermal conditions is not considering the
effect of salinity on water density. While it is usually a second order effect during summer stratification, when density is mostly
dependent on water temperature, the existence of a vertical variability of the salinity may become relatively more important
when the temperature is approximately uniform, with consequences on the freezing time in saline lakes (Stepanenko et al.,
2019). Adding this factor in the minimal model is possible, but would require the inclusion of a sub-model for the salinity
profile and of additional data that are not routinely available.

    Finally, we note that the model requires the estimate of a single parameter, the global efficiency $\eta$. Obtaining an accurate
value of this parameter based on theoretical considerations is a difficult task (see the discussion in the Supplementary Material
for some hints) and would require a much deeper hydro- and aero-dynamic analysis. However, it is clear that the calibration
of one parameter is not particularly challenging and can be pursued even if the available data are relatively limited. It is also
important to recognize that the time step $\Delta t$ plays a role both in the definition of the well-mixed layer thickness (through $E_w$)
and in the quantification of the heat loss $E_c$; the choice of a daily time step is the most appropriate choice because it integrates
the main periodicity of the external forcing.

**5.3  Implications**

The minimal model was designed to provide a simple process-based tool to estimate freezing dates in lakes. SELF can be
easily generalized at global scale as an operational prognostic product, as it relies on easily accessible quantities: surface heat
fluxes (which can be computed using variables from nearby meteorological stations or from global meteorological models, for
instance) and date for homothermal lake temperature. The necessary occurrence of homothermal conditions at the temperature
of maximum density (4°C), which can be detected from remotely sensed LSWT, can provide accurate initial conditions to
model the energy competition with SELF.

    A follow-up study should aim at using global meteorological data and remotely sensed temperature measurements from
satellites to predict the timing of ice formation, and potentially contributing in the monitoring of this essential climate variable,
as defined by GCOS (https://gcos.wmo.int/en/essential-climate-variables/lakes/, last access 26/04/2021). Note that the only
tuning parameter from SELF can be calibrated for each lake based on satellite-based observations of $n_d$. As mentioned above,
the homothermal conditions can be probed with satellite (infrared) optical radiometer and ice formation can be operationally
tracked with either optical or microwave remote sensing technique (Duguay et al., 2014). There is finally a practical interest



of SELF for lake managers as the model can be used to provide a short-term probability of the timing for ice formation. This kind of information may help stakeholders effectively face the strong inter-annual variability in ice phenology.

## 6 Conclusions

We developed a minimal model, SELF, to predict the duration of the pre-freezing period ranging from the early winter lake's overturn to the formation of an ice sheet at the surface. We showed that the temporal evolution of the thermal structure during this period is governed by the competition between cooling of the surface water due to the heat lost to the atmosphere and mixing of the surface layer due to wind. We demonstrated that including only those two physical processes in SELF is sufficient

to describe the first order dynamics of the inverse stratification process with only one calibration parameter. An approximate analytical solution obtained by further simplifying the minimal model in the ideal case of constant mechanical and thermal energy input can be used to sketch the general tendency of the system, highlighting the approximate power-law dependence on the energy fluxes, and eventually replacing traditional integral approaches such as negative degree days.

The simplicity of the model allowed us to perform Monte Carlo simulations and characterize the process as a function of the

cumulated or daily averaged values of the energy fluxes in statistical terms. Such analysis showed that the history of the system (i.e., the actual sequence of the atmospheric forcing) is crucial to determine the duration of the pre-freezing period exactly, but a general tendency can be recognized. We suggest that this competition between wind and heat loss could partly explain the strong inter-annual variability observed in the ice-on phenology worldwide.

In this work, we have focused on the mechanistic definition of the minimal model SELF with a validation restricted to alpine

lakes. Now we encourage two immediate applications of SELF. First, this model can be used at global scale to help understanding change in ice phenology. Second, the model could be used to help stakeholders evaluate the short-term probability of ice formation on their lakes.

*Code availability.* The source code of SELF (DOI: 10.5281/zenodo.5082374) is available at https://github.com/marcotoffolon/SELF.

*Author contributions.* Designing of the research: DB, MT. Performing the research: MT, DB, LC. Writing the manuscript: MT, DB, LC.

*Competing interests.* The authors declare that there is no conflict of interest regarding the publication of this article.

*Acknowledgements.* The authors would like to thank Nathalie Dubois (Eawag) for collecting and providing lake temperature data in Lac de Joux; Jean-Daniel Meylan (fisherman) for his long-term record of ice phenology in Lac de Joux; Michael Plüss (Eawag) and Felix Keller



(Academia Engiadina) for lake temperature and ice measurements in Engadin's lake. Hugo Ulloa is thanked for discussion on the paper. DB acknowledges the support from GCOS Switzerland and Academia Engiadina.





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
