# Peer review of "SELF v1.0: A minimal physical model for predicting time of freeze-up in lakes"

_Geoscientific Model Development, 2021_

## Author Response (AR1)

Response to Editor and Reviewers about "SELF v1.0: A minimal physical model for predicting time of freeze-up in lakes" by Marco Toffolon et al., Geosci. Model Dev. Discuss., https://doi.org/10.5194/gmd-2021-234-EC1, 2021

**Invitation to proceed with response and revisions**
Andrew Wickert (Editor)

Following the initial positive manuscript reviews, I would invite the authors to begin preparing their response to the referee comments as well as revisions to their article.

We thank the Editor for having handled our submission and for his appreciation of our work. We have replied to the Referees' comments in detail, and we are ready to submit the revised manuscript.

**Comment on gmd-2021-234**
Reply to Anonymous Referee #1

AR#1 - The authors propose a simple model of the seasonal lake cooling leading to formation of the ice cover. The model is calibrated and tested on data from alpine lakes and on outcomes of a one-dimensional process-based model. The model is of potential use in research on the lake ice cover formation and in lake-related applications. The fivefold variability of the main tunable parameter \eta between the five tested lakes (Table 2) raises however concerns about applicability of the model on large spatial scales. Another point of criticism is the description of model hidden largely in the supplemental, which is rather uncommon for a modeling journal and distracts the reader. Otherwise, the study is well conceived, applies adequate methodology and clearly written. Below are a couple of minor technical remarks.

We thank the Reviewer for the favourable general comments. Concerning the two elements of criticism, we provide specific replies in the following points.

1) We agree with the Reviewer that the variability of the parameter $\eta$ does not allow for imposing the same value for all cases. However, we do not expect $\eta$ to be a universal constant because it includes two factors that are intrinsically varying from lake to lake: the efficiency of energy transfer from wind stress to internal mixing; and the representativeness of the meteorological station. Our preliminary analysis suggests that the former factor might be affected by lake depth (as we note in L302, "with higher values [of $\eta$] for shallow lakes … and lower values for the deepest lakes …"), but it is not possible to obtain a reliable dependence with only five values, nor to highlight the dependence on other physical variables. In any case, the latter factor (how meaningful is the measured wind speed for the actual conditions on the lake) may jeopardize any attempt in that direction, and confirm the need to find an ad hoc value of $\eta$ for each pair of lake and meteorological station.
Nevertheless, $\eta$ is the only parameter that needs to be tuned in the model. If a sufficient long time record of measurements is available, the calibration is not a difficult task.

2) Regarding the structure of the paper and the presentation of the details of the model, we believe that the essential information is already in the main text, where all the physical arguments are explained. The reader, after having had an overview of the model features, can rapidly go into the details provided in the Supplementary Material. We also want to stress that the discussion in the SM is not concerned with the SELF model, but with the theoretical background, the physical interpretation, and the derivation of the approximate analytical solution.

**AR#1 - L42: The description of the models cited in this paragraph needs refinement. ``Statistical approach'' is not a proper definition here; the model of Rhode (1952) and Billelo (1964) is not statistical. Leppäranta did not propose any model of ice formation in his 1993 paper.**

The Reviewer is right. The models are based on a linearization of the heat fluxes between the atmosphere and the lake leading to a first order differential equation. We have removed the word "statistical" and restructured the paragraph. We also now refer to the work from Leppäranta (2014).

**AR#1 - L168: "The model SELF does not admit an analytical solution in closed formed." – the sentence is unclear. Does it mean the equation is not solvable in quadratures? Reformulate in a clearer way.**

We revised the sentence as: "The set of equations that composes the model SELF does not admit an analytical solution in explicit form, for instance in terms of a relation for the number of days $n_d$ as a function of the forcing."

**AR#1 - L238: "a.s.l. (above sea level)" - replace with "above sea level (a.s.l.)"**

Corrected.

**AR#1 - L249, L286, Fig. 3 and elsewhere: Replace "performances" with "performance".**

Corrected.

Additional references:
Leppäranta M. (2014). Interpretation of statistics of lake ice time series for climate variability. *Hydrology Research* 45 (4-5): 673–683. doi: 10.2166/nh.2013.246

**Comment on gmd-2021-234**
Reply to Anonymous Referee #2

**AR#2 - The authors have developed an excellent minimal process-based model to project the timing of ice on in lakes. By showing how it gives similar results to a complex vertical 1D hydrodynamic model and outperforms popular empirical approaches (negative degree days), the authors give a convincing argument that the methodology should be applied to global lake studies. The language of the manuscript is very good, and the methodology is well laid out. Although some supplementary information could arguably have been part of the main text, the methods and considerations are clear and described in detail. Especially the pseudo analytical solution of the model framework using MCMC could provide the basis for future ice phenology studies. This manuscript is clearly suited for GMD and should be published soon.**

We warmly thank the Reviewer for his/her appreciation of our work and for the detailed suggestions. We do hope that the model can be useful to the scientific community for future studies.

**AR#2 - Major comment:**
**AR#2 - L363-368: I couldn't follow the argument here that because the importance of wind increases with lower values for heat loss and due to global warming, we would expect more year-to-year variability in ice-on timings, but we still can't use lake ice phenology as sentinel for climate change. Why is that? Wouldn't the decreasing heat losses due to global warming make ice phenology a prime example for an example of the consequences of global warming? Or is the argument here that changes in the wind field are not primarily influenced by climate change?**

We fully acknowledge that ice phenology can be considered as a valuable sentinel for climate change; moreover, it is an indicator that can be assessed in a simple way by means of visual observation of the ice cover. However, climate change interacts with ice phenology in different ways. First, a climate becoming warmer typically postpones the day of homothermal conditions (when our model SELF starts). Second, it modifies the duration of the pre-freezing period: warmer air temperature tends to delay the day of ice formation, as well. Yet, we demonstrate that the process is also affected by the wind energy transferred to mixing the surface layer: weaker/stronger winds anticipate/postpone ice formation.

Therefore, we should compare the interannual variability of ice phenology with the interannual variability of air temperature and wind speed. If the interannual variability of ice phenology becomes larger than, for instance, that of air temperature due to the effect of the wind, then, ice phenology can become a confusing signal of climate change. In this respect, we found that the influence of wind is significant for warm conditions (e.g., lower altitude), and that it might overcome the direct effect of climate warming if the change of wind speed is sufficiently important. Our comment was focused on this.

Said differently, the continuous delay in the day of ice formation is a clear signal of climate warming at high latitude/altitude, but the signal is perturbed at places where the temperatures do not drop significantly below 0°C anymore in winter. In such places, the stochasticity of the other meteorological forcing, and notably the wind, can play a non-negligible role leading to an increase in the variability of ice phenology. Along these lines, we revised the whole paragraph as follows.

*"SELF can be used to better understand long time series of ice formation in lakes (Magnuson 2000) and specifically to decouple the inter-annual variability from the long-term climate-change induced trend. A first effect of warming, which we do not discuss here, is the delay of the day when homothermal conditions occur. A second effect is the modification of the duration of the pre-freezing period. In this respect, the analysis based on random sequences suggests that the influence of wind increases for warm climates (low latitude/altitude), and that this effect might become relevant if wind change is sufficiently strong. If the inter-annual variability of ice phenology becomes larger than, for instance, that of air temperature due to the effect of the wind, then, ice phenology might become a confusing signal for climate change."*

AR#2 - Minor comments:
AR#2 - L71: The phrase "cooling progressively diffuses downwards" is a bit vague to me, as cooling as a heat loss flux should not diffuse. The thinking behind the statement is clear that cooling causes stratified conditions, but isn't this more related to the closeness of the atmospheric boundary condition? Further "warming below 4 deg C" is also confusing in my opinion. Is the argument here that if the surface layer warms up closer to 4 deg C, this will cause convective overturn with lower layers that are less than 4 deg C? It would be good to make these two sentences clearer

The Reviewer is right, the two sentences were not accurately expressed. We revised them as follows:
"When heat is lost, the surface layer stably stratifies because water cools progressively from the free surface downwards. When water at the surface warms, but remains below 4°C, instead, the stratification is unstable due to convective overturn and is readily mixed in the subsequent phase A."

AR#2 - L74: the text switches between describing the curves as concave or parabolic, maybe sticking to one would be easier for the reader

We changed 'parabolic' into 'concave' to refer to the same concept with the same word. This also avoids confusion with 'parabolic' (2nd order) approximation of the equation of state.

AR#2 - L85: W for the wind speed wasn't introduced yet and should be defined here

We thank the Reviewer for having noting it. We have added the explicit definition "wind speed $W$" in the sentence at L85.

AR#2 - L96: Shouldn't this be "(ii) the final temperature profile in the newly created mixed layer after cooling caused stabilizing conditions (phase B)", or something similar?

We followed the Reviewer's suggestion and we revised the text accordingly.

AR#2 - L107: I think it would be good to state that the (potential) energy is given in J/m2 as otherwise some readers would be confused why you don't integrate over A(z)

Thanks. We changed it as "potential energy per unit area [J m$^{-2}$]".

AR#2 - Equation 6: It would be good to state in a line before that Ec = rho cp h delta T / 2 (supplement S.18) to help the user to get the step to eq. (6)

We agree with the suggestion. We modified the sentence as "Given the thickness $h$ of the surface layer and the heat loss $E_c$, the assumed linear temperature profile establishes a relation $E_c = \rho_0 c_p h \Delta T/2$, where $\rho_0$ is a reference value for water density. Thus, it is possible to compute […]"

AR#2 - L168: It should be "in closed form" here

Following Reviewer #1's suggestion, we revised the whole sentence as "The set of equations that defines the model SELF does not admit an analytical solution in explicit form, for instance in terms of a relation for the number of days $n_d$ as a function of the forcing variables.

AR#2 - 4: I really enjoyed reading that you argued the discrepancies between the two process-based models and the logger data is due to shortcomings of the meteorological driver data, and I totally agree with that statement

We are glad that the Reviewer shares our concern on this issue.

AR#2 - L299: I would exchange "surprising" with "promising"

Revised as suggested.